# Cardiorespiratory Fitness in Low Calcium Consumers: Potential Impact of Calcium Intake on Cardiorespiratory Fitness

**DOI:** 10.3390/nu17193138

**Published:** 2025-09-30

**Authors:** Julian Kennedy, Louis Pérusse, Vicky Drapeau, Angelo Tremblay

**Affiliations:** 1Department of Kinesiology, Université Laval, Québec, QC G1V 0A6, Canada; julian.kennedy.1@ulaval.ca (J.K.); louis.perusse@kin.ulaval.ca (L.P.); vicky.drapeau@kin.ulaval.ca (V.D.); 2Centre Nutrition, Santé et Société (NUTRISS), Institut sur la Nutrition et les Aliments Fonctionnels (INAF), Université Laval, Québec, QC G1V 0A6, Canada; 3Quebec Heart and Lung Institute Research Center, Université Laval, Québec, QC G1V 0A6, Canada

**Keywords:** aerobic performance, body composition, oxygen consumption

## Abstract

**Background:** Calcium is essential for maintaining bone health, facilitating muscle contractions, regulating body temperature, and supporting aerobic metabolism. While the relationship between physical activity and calcium metabolism is well established, the impact of calcium intake on cardiorespiratory fitness (CRF) remains underexplored. The main aim of this study was to assess the effects of calcium intake on CRF and the mediation effect of calcium intake on the relationship between vigorous physical activity participation and CRF. **Methods:** Analyses were performed on a sample of 576 adult participants (257 males and 319 females) from the Quebec Family Study (QFS) for whom data were available for calcium intake, CRF, and body composition. The effects of calcium intake on CRF and body composition were analyzed by comparing subjects classified into sex-specific tertiles of calcium intake using general linear mixed models. Pearson correlations were also used to document the associations between calcium intake, CRF, and body composition. A mediation analysis was used to determine the effect of calcium intake as a mediator of the association between vigorous physical activity and CRF. **Results:** The comparison of calcium-based tertiles revealed that low calcium consumers had lower CRF, especially in males. In both males and females, correlation analysis showed that calcium intake was positively associated (*p* < 0.05) with CRF. Mediation analyses revealed that calcium intake explains an insignificant fraction of the physical activity–CRF relationship. Between-tertile comparisons showed reduced body fat and increased fat-free mass levels when increasing calcium intake, although some of these effects were not statistically significant. **Conclusions:** The results of this study suggest that low calcium intake is associated with reduced aerobic capacity in adult males and females. While the positive relationship observed between calcium and aerobic fitness aligns with calcium’s known physiological roles, further research is needed to clarify the exact mechanisms by which this micronutrient may influence aerobic capacity.

## 1. Introduction

Over the past decades, research has shown that calcium plays critical roles in maintaining optimal physiological functioning and overall health. Calcium is essential for bone health, muscle contraction, nerve impulse transmission, and energy metabolism [1]. Despite its importance, inadequate calcium intake remains a common dietary issue. In Canada, over 40% of males and 60% of females did not meet their recommended dietary calcium intake in 2021 [2]. Such deficiencies are associated with reduced bone mineral density and an increased risk of osteoporosis [3], and emerging evidence links them to cardiometabolic disorders such as obesity, type 2 diabetes, and dyslipidemia [4].

Fundamental physiological mechanisms underscore the role of calcium in exercise performance. As mentioned above, calcium is crucial for muscle contraction, nerve signal transmission, and mitochondrial ATP production [5,6,7]. It also plays a key role in thermoregulation during physical exertion by stimulating sweat secretion and regulating its hypotonicity [8]. Furthermore, vitamin D, which facilitates calcium absorption and metabolism, must also be considered. Deficiencies in vitamin D frequently co-occur with low calcium intake and have been associated with diminished muscle strength and reduced aerobic capacity [9].

Calcium homeostasis is fundamental to the functioning of all cell types, which is why the levels of calcium in the blood are tightly regulated within a very narrow range. Balance is maintained through the coordinated regulation of intestinal absorption, renal excretion, and skeletal storage. Parathyroid hormone (PTH) and other hormones play a critical role in this process. Serum calcium levels govern PTH secretion via a negative feedback mechanism mediated by the calcium-sensing receptor (CASR) in the parathyroid glands. Once released, PTH acts rapidly to increase renal calcium reabsorption, thereby helping to restore homeostatic levels [10].

While the relationship between physical activity and calcium metabolism has been well documented with respect to bone mineral density, the impact of calcium intake on aerobic performance remains underexplored in the scientific literature. Interestingly, this topic is more frequently discussed in coaching literature, where concerns about calcium’s role in performance are common. In this regard, Peacock et al. [11] reported an improvement induced by a single dose mineral supplement containing calcium on maximal oxygen consumption (VO_2_ max) and the perceived exertion index. However, their results did not allow for determining the specific contribution of calcium to these effects.

Although the precise mechanisms linking calcium to aerobic performance are not yet fully understood, hormonal adaptations may offer some explanation. For instance, elevated PTH levels in response to low calcium intake can promote calcium accumulation in soft tissues [12]. In adipocytes, increased intracellular calcium favors lipogenesis over lipolysis, potentially reducing whole-body fat oxidation [13]. Similar mechanisms could adversely affect muscle cells, possibly leading to mitochondrial inefficiency or, in extreme cases, apoptosis [14]. Additionally, PTH may impair muscle performance by inhibiting carnitine palmitoyl transferase (CPT), thereby reducing the muscle’s capacity for ATP production via β-oxidation [15].

Assessing the link between calcium intake and CRF helps to better understand how dietary calcium influences aerobic performance. Insights into dietary calcium can contribute to the development of evidence-based nutritional guidelines for athletes and active individuals, ultimately promoting enhanced performance and overall health. This study examines the correlation between calcium intake, aerobic performance, and physical working capacity using data from the Quebec Family Study (QFS). We hypothesized that a higher intake of both calcium and vitamin D should correlate positively with CRF. To further document the influence of calcium on these associations, we evaluated the mediation role of calcium intake on the relationship between CRF and vigorous physical activity participation.

## 2. Methods

### 2.1. Participants

The subjects of the present study were participants in the Quebec Family Study (QFS) described in detail elsewhere [16,17]. Briefly, the QFS is an observational study with three phases of data collection that took place between 1979 and 2002 and aimed to investigate the role of genetics in the etiology of obesity and related cardiovascular risk factors. During Phase 1 of the study (1979 to 1982), a diverse cohort of 1630 individuals from 375 families with a broad range of body mass indices (BMIs) from 13.8 to 64.9 kg/m^2^ was recruited. Recruitment during this phase was inclusive, with no consideration of body weight. In Phases 2 (1989–1997) and 3 (1998–2002), families with at least one parent and one child with a BMI of 32 kg/m^2^ or higher were added to the cohort. All participants were French Canadians from the greater Quebec City area and were recruited through various media outlets. They were volunteers and gave their written consent to participate in this study, which was approved by the Medical Ethics Committee of Laval University [17]. The present study includes 576 participants (257 males and 319 females) tested in Phases 2 or 3 with complete cross-sectional data for calcium intake, CRF, and body composition.

### 2.2. Dietary Record

Food intake was determined by using a 3-day dietary record, as previously described [18]. The calcium and vitamin D contents of the diet were calculated with the Canadian Nutrient File. The dietary journal was completed on 2 weekdays and 1 weekend day.

### 2.3. Physical Working Capacity

The cardiorespiratory fitness (CRF) of the participants was assessed using a submaximal exercise test performed on a cycle ergometer, as previously described [19]. The test included three 6-min workloads, each separated by a 1-min rest. The physical working capacity of the participants, defined as the power output at a heart rate of 150 beats per minute (PWC150), served as an indicator of CRF. This test also allowed for the estimation of VO_2_ max by regression analysis from the VO_2_ at the end of each work load and the corresponding heart rate to maximal heart rate adjusted for age [20].

### 2.4. Physical Activity Participation

Participants completed a physical activity record for 3 days [21]. This record was completed in the same time frame as the dietary record, including 2 weekdays and 1 weekend day. Each day was split into 96 periods of 15 min, and participants were instructed to code the main activity of each period on a scale from 1 to 9 corresponding to increasing levels of energy expenditure from 1 MET (resting) to 9 METS (intense physical activity). The number of periods coded at 8 and 9 over the 3 days (MN89) was considered as an indicator of vigorous physical activity. This record was shown to provide highly reproducible results [21].

### 2.5. Anthropometric and Body Composition Measurements

Several variables were measured from the participants. Height was measured to the nearest 0.1 cm using a standard stadiometer, while body weight was recorded to the nearest 0.1 kg using a digital panel indicator scale (Beckman Industrial Ltd., Model 610/612, Scotland, UK). The body mass index (BMI) was calculated using these measurements. Waist circumference was measured above the uppermost lateral border of the iliac crest as previously described [22]. Body density was assessed by hydrodensitometry. The percentage of body fat was estimated from body density using the Siri formula [23]. Fat mass was calculated from the derived percentage of body fat and total body weight, while fat-free mass was calculated by subtracting FM from total mass.

## 3. Statistical Analysis

Pearson correlations were calculated to assess the age-adjusted associations between nutrient intake, CRF, and morphological indicators. The effects of calcium intake on CRF and body composition were determined using general linear mixed models implemented in the MIXED procedure of SAS (SAS University Edition, version Oct 2019), whereas the Tukey HSD test was used for a posteriori comparison. Analyses were performed separately in males and females by comparing subjects classified into tertiles of calcium intake. Mediation analysis was also used to determine the role of calcium intake as a mediator of the relationship between vigorous physical activity and CRF. As previously described in detail elsewhere [16], mediation analysis is a tool used to assess the extent to which the effect of an independent variable X (here, vigorous physical activity) on a dependent variable Y (here, CRF) could be explained by an intermediate variable called a mediator. Traditional mediation analysis decomposes the total effect of X on Y into direct and indirect effects, that is, the effect that is not via the mediator versus the effect that is via the mediator, respectively. We used a tool called PROCESS to assess the total, direct, and indirect effects and to calculate the percentage of mediation as the ratio of the indirect effect to total effect [24].

## 4. Results

Table 1 shows that there were large differences in daily calcium intake between tertiles of this nutrient in males and females. An adequate intake was only observed in the high tertile, whereas it was insufficient in the other two tertiles for both sexes. This table also highlights substantial variations in age, which justified a statistical adjustment for this variable in the comparison of CRF and body composition values between calcium tertiles. The results of these comparisons in males and females are presented in Table 2a and b, respectively. As expected based on the known physiological properties of calcium, CRF was lower in low calcium consumers, especially in males. The same trend was found in females, but the between-tertile difference did not reach standard statistical significance. These tables also show that there was no between-tertile difference in body weight and composition indicators, except for percent body fat in females, which was lower in the high than in the low calcium consumers.

Table 3 shows the correlation between calcium and vitamin D intake with age, aerobic fitness, participation in vigorous exercise weekly, and body composition. In males and females, there was a significant negative association between calcium intake and age. In both sexes, calcium intake was positively correlated with PWC150 and VO_2_ max. Calcium intake was also significantly related to participation in vigorous physical activity in males, whereas this association did not reach statistical significance in females. Calcium intake was associated with a reduction in percent body fat and fat mass in males. The same trend was found in females, although statistical significance was only observed for percent body fat. There was no relationship between vitamin D and fitness or body composition in both sexes. In males, vitamin D intake was positively correlated with participation in vigorous physical activity.

Mediation analysis was also performed to better understand the relationship between calcium intake and CRF. Considering the positive relationship between calcium intake, physical activity, and CRF, the mediation analysis was performed to determine if calcium intake is a mediator of the relationship between physical activity and CRF. The analysis revealed that the percentage of the total effect of vigorous physical activity participation on CRF that is attributable to the mediation by calcium is about 5% and is not statistically significant.

## 5. Discussion

The primary finding of this study is that an inadequate calcium intake is associated with reduced CRF. Conversely, there was no association between CRF and vitamin D intake. This finding suggests that supplemental calcium intake has little impact on aerobic fitness in individuals with adequate to high consumption levels, but positive effects may be seen in those with a lower intake.

The potential influence of variations in calcium intake on CRF is not clearly understood, although the available literature suggests several potential explanations. In a study conducted by Peacock et al., a single dose of calcium lactate, magnesium lactate dehydrate, and zinc oxide supplement before exercise had a significant effect on VO_2_ max [11]. Additionally, Greer et al. found a positive correlation between whole-body mineral density (BMD) and VO_2_ max, suggesting that higher BMD may be associated with better aerobic performance [25]. Since bones are the primary storage site for calcium in the body, a higher BMD could indicate greater calcium bioavailability. Calcium plays a crucial role in muscle contraction and energy metabolism, both of which are vital for maintaining high levels of aerobic capacity and endurance. With greater calcium reserves in the body, muscle function could be enhanced, leading to improved VO_2_ max performance. Therefore, individuals with higher bone mineral density may have more accessible calcium for muscle contraction, potentially supporting better cardiovascular and muscular efficiency during intense exercise, which, in turn, could contribute to higher VO_2_ max values. Taken together with these studies, the results of the present study show a significant positive correlation between aerobic fitness and calcium consumption. When our participants were subdivided into tertiles based on their daily calcium intake, it seems that this relationship mainly reflects the influence of low calcium intake, since these consumers displayed lower levels of aerobic fitness, whether it was measured with PWC150 or the estimated VO_2_ max.

Studies such as those by Zemel [26] have shown that low dietary calcium leads to increased intracellular calcium via elevated calcitriol levels in adipocytes, a mechanism that may extend to muscle cells. If skeletal muscle cells respond similarly, calcium overload could impair mitochondrial function and cell viability. Low calcium intake also prompts a compensatory rise in parathyroid hormone (PTH), which further contributes to muscle dysfunction. Elevated PTH alters calcium distribution within soft tissues and has been shown to inhibit carnitine palmitoyl transferase (CPT), a key mitochondrial enzyme responsible for fatty acid oxidation, according to studies by Smogorzewski et al. [15] and Perna et al. [27]. This inhibition reduces the muscle’s ability to generate ATP via β-oxidation, increasing reliance on limited glycogen stores and accelerating fatigue during aerobic exercise. These effects can be partially reversed by calcium channel blockers such as verapamil, underscoring the detrimental role of dysregulated calcium metabolism on mitochondrial efficiency. In another study by Li et al. [28], mitochondrial calcium homeostasis was found to be vital for aerobic metabolism. Excess calcium within mitochondria initially activates oxidative phosphorylation but, when unregulated, leads to oxidative stress, opening of the mitochondrial permeability transition pore (MPTP), and activation of cell death pathways. These mitochondrial disruptions impair energy production and reduce endurance, directly limiting aerobic performance [28].

Mediation analysis was also used in this study because in addition to the natural association between vigorous physical activity participation and CRF, we observed an association between calcium intake and these two variables. As indicated above, calcium intake did not appear to be a mediator of the activity–fitness relationship as the indirect effect was not significant. This suggests that the relationship between exercise and CRF is not primarily explained by variations in calcium intake. As described above, the available evidence rather suggests that insufficient calcium intake promotes metabolic changes that ultimately affect aerobic performance and CRF.

Beyond metabolic effects, PTH appears to have direct muscular consequences even under conditions of adequate calcium levels. Voss et al. [29] found reduced muscle strength in both normocalcemic and hypercalcemic primary hyperparathyroid (PHPT) patients, suggesting that elevated PTH, independent of serum calcium, impairs muscular function. These deficits were improved following parathyroidectomy in multiple studies [30,31,32], supporting the conclusion that elevated PTH compromises muscle performance and is reversible with hormonal normalization. Diniz et al. [33] further demonstrated that PTH may contribute to neuromuscular dysfunction, with altered nerve conduction and reduced motor unit efficiency. These biological mechanisms could be the possible link for our observational findings. When our participants were subdivided by calcium intake levels, those in the lowest tertile exhibited reduced aerobic fitness, suggesting that insufficient calcium availability contributes to poor muscle performance and energy metabolism. While our study did not detect a direct effect of vitamin D intake on aerobic capacity, other studies have reported positive associations [34], potentially via vitamin D receptor activation in muscle cells that promotes protein synthesis and tissue repair [35,36]. Through these mechanisms, adequate vitamin D levels could support muscle function and recovery, thereby potentially enhancing aerobic capacity and overall physical performance. However, inconsistencies in the literature highlight the need for further research to clarify the specific impact of vitamin D intake on aerobic fitness across different populations.

Even if this study was not primarily performed to examine sex differences in the relationship between calcium intake and aerobic fitness, it is worth emphasizing that, to our knowledge, this is the first study documenting the relationship between males and females. As described above, a positive relationship between these two variables was observed in both sexes, although the statistical significance of the relationship between these two variables was more pronounced in males. This may possibly be explained by the relatively higher fat-free mass in males. Since calcium intake and CRF values are both greater in males, this offers a wider range of variance contributing to better statistical discrimination in males.

Increased calcium consumption offers numerous benefits, such as supporting bone health, improving muscle function, or potentially aiding in aerobic fitness as seen in this study. However, it is important to approach calcium intake cautiously, as there are ongoing debates regarding potential risks. One concern is that calcium supplements may increase the risk of cardiovascular disease [37]. Despite these concerns, it remains unclear whether the method of calcium intake, whether through supplementation or natural sources such as dairy products, has a significant impact on heart health. Additionally, the specific amount of daily calcium consumption at which these detrimental effects begin to emerge is still not well established. Therefore, while the benefits of calcium are clear, further research is needed to better understand the associated risks and to guide safe and effective calcium consumption practices. In this regard, the results of the present study suggest that the benefit of calcium supplementation might be mainly observed in low calcium consumers, thus making the supplementation less relevant in consumers with adequate intake. Therefore, the debate surrounding the potentially detrimental effects of high calcium intake on the risk of cardiovascular disease does not invalidate the main clinical implication of the present study, highlighting the relevance of preventing calcium deficiency to maintain optimal aerobic fitness.

As expected, an insufficient calcium intake was associated with a body composition profile with higher fat mass [38,39]. The available literature suggests that the increased tendency of the low calcium consumer towards body fat accumulation is explained by a decrease in fat oxidation [13,40] and an increase in energy intake [41]. As described above, it is plausible that these effects are mediated by variations in calcemia and its related hormonal regulation.

This study has several limitations. Firstly, the majority of the participants were Caucasian, which limits the generalizability of the results to other demographic groups. Secondly, blood calcium levels were not directly measured during the exercise protocol, potentially affecting the accuracy of conclusions related to calcium metabolism. Additionally, as this study was transversal, there was no follow-up with the participants to observe the different impacts of calcemia on their CRF variables over time. This limits the study’s generalizability. These factors could have influenced this study’s findings and should be considered when interpreting the results.

This study also has some strengths, which include the fact that aerobic capacity was measured under well-standardized conditions during the cycling exercise test, allowing for the measurement of the amount of work performed at a heart rate of 150 BPM. Another strength of this study is the high statistical power of the analysis that was conferred by the large number of participants of both sexes in the Quebec Family Study.

## 6. Conclusions

In summary, the results of the present study show that a low calcium intake is associated with a reduced CRF in adult males and females. The same trend was not observed when analyzing the relationship between vitamin D intake and CRF. Even if the observation of a positive calcium–CRF association is concordant with the documented functions of calcium, further research is needed to describe the mechanisms underlying the potential impact of the mineral on aerobic fitness as well as its implications for performance and health.

## Figures and Tables

**Table 1 nutrients-17-03138-t001:** Age and calcium intake in males and females by tertiles of calcium intake.

Variable	Tertiles of Calcium Intake		
**Males**	Low (n = 85)	Middle (n = 86)	High (n = 86)
Age (years)	43.7 ± 14.1	40.8 ± 15.0	32.9 ± 13.2
Calcium intake (mg/day)	579 ± 140	980 ± 134	1544 ± 314
**Females**	Low (n = 106)	Middle (n = 107)	High (n = 106)
Age (years)	40.3 ± 14.2	37.5 ± 13.2	34.9 ± 13.4
Calcium intake (mg/day)	500 ± 105	818 ± 78	1214 ± 271

Values are means ± SD.

**Table 2 nutrients-17-03138-t002:** (**a**). Fitness and anthropometric measurements in males by tertiles of calcium (Ca). (**b**). Fitness and anthropometric measurements in females by tertiles of calcium.

(a) Males											
Variable	Low Ca			Middle Ca			High Ca			*p* Value	Group Difference
	N	Mean	SEM	N	Mean	SEM	N	Mean	SEM		
PWC150/KG	78	10.76	0.38	76	10.65	0.38	83	11.99	0.38	0.0239	H-M
MN89	81	1.62	0.37	85	1.36	0.36	83	2.48	0.38	0.0891	
VO_2_ max (mlO_2_/kg.min^−1^)	85	25.43	0.75	86	25	0.75	86	28.13	0.77	0.0086	H-M/H-L
VITAMIN D (mcg/d)	85	4.44	0.43	86	6.53	0.43	86	9.89	0.44	<0.0001	H-L/H-M/M-L
BODY WEIGHT (cm)	85	78.86	1.81	86	76.63	1.8	86	80.37	1.88	0.2871	
BMI (kg/m^2^)	85	26.28	0.55	86	25.46	0.55	86	26.62	0.57	0.2398	
PERCENT BODY FAT	81	22.74	0.84	86	21.13	0.82	85	20.93	0.85	0.2198	
FAT MASS (kg)	81	19.42	1.16	86	17.12	1.13	85	17.73	1.18	0.2789	
FAT-FREE MASS (kg)	81	60.18	0.87	86	59.51	0.85	85	62.74	0.88	0.0157	H-M
(**b**) **Females**											
**Variable**	**Low Ca**			**Middle Ca**			**High Ca**			***p* Value**	**Group Difference**
	**N**	**Mean**	**SEM**	**N**	**Mean**	**SEM**	**N**	**Mean**	**SEM**		
PWC150/KG	82	6.72	0.26	90	7.22	0.25	87	7.53	0.26	0.0731	
MN89	99	0.86	0.18	104	0.87	0.18	102	1.38	0.18	0.0806	
VO_2_ max (mlO_2_/kg.min^−1^)	106	16.65	0.47	107	17.6	0.47	106	17.92	0.47	0.1184	
VITAMIN D (mcg/d)	106	2.91	0.26	107	5.12	0.26	106	7.37	0.26	<0.0001	H-L/H-M/M-L
BODY WEIGHT (cm)	106	69.14	1.75	107	68.47	1.76	106	66.95	1.78	0.629	
BMI (kg/m^2^)	106	26.92	0.67	107	26.28	0.67	106	25.73	0.67	0.4122	
PERCENT BODY FAT	95	33.15	0.96	104	30.55	0.92	104	29.49	0.93	0.0118	H-L
FAT MASS (kg)	95	24.33	1.31	104	22.54	1.26	104	21.01	1.27	0.1484	
FAT-FREE MASS (kg)	95	45.28	0.63	104	46.12	0.61	104	45.32	0.61	0.5011	

Values are means ± SEM after adjustment for age. PWC150/kg: Power work capacity at 150 heart beats expressed per kg of body weight. NM89: Vigorous physical activity participation expressed as the number of 15 min periods graded at 8 and 9 over 3 days. VO_2_ max: Estimated VO_2_ max at the age predicted maximal heart rate.

**Table 3 nutrients-17-03138-t003:** Correlations between nutrient intake, fitness, and anthropometric measurements.

Variable	Males		Females	
	Calcium	Vitamin D	Calcium	Vitamin D
Age	−0.34 ***	−0.04	−0.16 **	0.04
PWC150	0.16 *	0.12	0.15 *	0.02
VO_2_ max	0.31 ***	0.12	0.19 ***	0.01
MN89	0.14 *	0.14 *	0.10	0.01
BODY WEIGHT	−0.02	0.02	−0.03	0.09
BMI	−0.10	0.01	−0.09	0.09
PERCENT BODY FAT	−0.30 ***	−0.06	−0.17 **	0.01
FAT MASS	−0.21 ***	−0.05	−0.09	0.06
FAT-FREE MASS	0.22 ***	0.09	0.05	0.10

*p* < 0.05 *; *p* < 0.01 **; *p* < 0.001 ***.

## Data Availability

The original contributions presented in this study are included in the article. Further inquiries can be directed to the corresponding author.

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
