# Peer review of "Cardiorespiratory Fitness in Low Calcium Consumers: Potential Impact of Calcium Intake on Cardiorespiratory Fitness"

_nutrients, 2025, doi:10.3390/nu17193138_

Round 1
Reviewer 1 Report
Comments and Suggestions for Authors
Cardiorespiratory Fitness in Low Calcium Consumers
The aim of this study was to assess the effects of calcium intake on cardiorespiratory fitness and on the relationship between vigorous physical activity participation.
Main concerns
- Current title is very vague and not specific enough. Also, please consider replacing the word “consumer” – this word at first glance implied individuals who “purchased” rather than ingesting calcium! Also, the title should include the word “daily intake of calcium” – anything that similar to this.
- The term “body composition” or any other terms with regard to body composition was not even mentioned in the abstract. But when I read the main text and results and discussion, there was some emphasis on body fat and muscle mass I relation to calcium daily intake. Please rewrite the abstract.
- Line 66. I have a looked at the Peacock et al study. Firstly, I don’t think this study is a valid peer-reviewed journal publication. Secondly, the subjects in the study ingested not only calcium bit other forms of minerals - hence the point that calcium alone may improves aerobic performance as you have currently written and implied here, is clearly not so correct. I suggest please write-out in greater details of the findings of the Peacock et al study and also highlight its limitations. Further in Line 67. You wrote “though findings across studies remain inconsistent” – you must provide references or write out the findings of these several studies to support your point.
- Line 121 and 122. Were the participants in the present study taught clearly how to determine the level or intensity of their exercise or physical activity? Because if they do not know how to gauge properly – the validity of their ability to code their exercise or physical activity as “vigorous” is questionable.
- The difference between PWC150 and estimated VO2max (in the context of the present study’s aim) should be clearly explain to the reader earlier on in the Methods section – so they better understand why the relationship between daily Calcium intake with PWC150 is not equivalent or the same as to the relationship between daily calcium intake and estimated VO2max.
- Limitation of the Person Correlation should e reflected here in the Limitations of the study. We know that significant correlation is not “cause and effect” and that only a higher value of > 0.7 is deemed as some sort of “very strong” relationship.
Minor issues:
- Line 27-28. The value of Pearson correlation should be written here for the awareness of the reader.
- Line 50. Was Vit D measured in the present study? If not, the relevance of this sentence is questionable, and also this should highlighted as another major limitation of the present study.
- Line 55. Why include “phosphorus” here??
- Some sentences are incomplete. For example, Line 58 – what do you mean by “this process”?
- Some sentences are incomplete. Line 71 & 72. Tou wrote “For instance, elevated PTH levels in response to low calcium intake can promote calcium accumulation in 71 soft tissues [12].” – so what’s your point here - and what are trying to imply?
- Line 83. “OFS” is used the first time here and should be defined.
- Line 197. “idea’ is not a good word here. Please change. I was thinking of “belief or concept” but still not a good work though.
- Table 2a and Table 2b. al, tables should be stand-alone. “CA” should be defined. Also, it should be “estimated or predicted VO2max”.
- Line 201 and 202. Is the Peacock study an acute or chronic intervention? Details of the study should be highlighted as mentioned previously. This will allow the reader to better understands the context when comparing between Peacock study’s findings and its relevance to the present study’s results.
- Line 271. “higher” is better than “increase”?
- Delete the term, “From NLM Medicine” and the all article doi should have the internet link – written as https://doi.org/XXXXXXXX.
Reviewer 2 Report
Comments and Suggestions for Authors
This study aims to assess the effects of calcium intake on CRF and the mediation effect of calcium intake on the relationship between vigorous physical activity participation and CRF. I think some things need clarifying for the publication that will help in the overall interpretation and understanding of the results before being published within the scope of the Nutrients.
Introduction
Comment 1: Informative and clear
Material, Methods and Results
Comment 2: Please, describe the assessment protocol and corresponding tool for bone density. Add reference.
Comment 3: Please add the Siri Formula and reference.
Comment 4: To improve the manuscript and facilitate readers' understanding, I suggest that the authors draw an explanatory figure of the mediation model.
Comment 5: Please add the bone density data to all tables.
Comment 6: line 271 -272: the authors wrote “Since calcium intake and CRF….. in males.” I am not sure about this. The analyses do not consider other variables that may be confounding factors (as shown in the correlation table). The authors should present their conclusions using different regression models to truly identify the relationship between the variables of interest.
Round 2
Reviewer 1 Report
Comments and Suggestions for Authors
Thank you for making the changes.
Reviewer 2 Report
Comments and Suggestions for Authors
Comment: The authors demonstrate several correlations between the variables analysed. In Table 3, the authors identify a positive correlation between VO2 max and calcium adjusted for age in both sexes for active participants (+150’). However, in the text following the table, they identify that there is no mediation between physical activity and VO2 max and therefore do not present the results. Based on the title ‘Cardiorespiratory Fitness in Low Calcium Consumers,’ it would be important for readers and the scientific community to understand the factors that influence CRF in low calcium consumers, as well as in other calcium consumer clusters. Is the BP variable an adjustment variable or an indicator of VO2 max? Thus, I suggest that the authors analyse the VO2 max indicators in the different calcium consumer clusters and compare any differences between the results of each cluster with adjustment variables. The discussion should be conducted in light of the new analyses.
The analyses presented in the current document are insufficient and do not allow such conclusions to be drawn: ‘The results of this study suggest that low calcium intake is associated with reduced aerobic capacity in adult males and females.’